# PIMT Controls Insulin Synthesis and Secretion through PDX1

**DOI:** 10.3390/ijms24098084

**Published:** 2023-04-29

**Authors:** Rahul Sharma, Sujay K. Maity, Partha Chakrabarti, Madhumohan R. Katika, Satyamoorthy Kapettu, Kishore V. L. Parsa, Parimal Misra

**Affiliations:** 1Center for Innovation in Molecular and Pharmaceutical Sciences (CIMPS), Dr. Reddy’s Institute of Life Sciences (DRILS), University of Hyderabad Campus, Hyderabad 500046, India; 2Division of Cell Biology and Physiology, Council of Scientific and Industrial Research-Indian Institute of Chemical Biology, Kolkata 700032, India; 3Central Research Lab Mobile Virology Research & Diagnostics BSL3 Lab, ESIC Medical College and Hospital, Hyderabad 500038, India; 4Department of Cell and Molecular Biology, Manipal School of Life Sciences, Manipal Academy of Higher Education (MAHE), Manipal 576104, India

**Keywords:** PIMT, T2DM, islets, beta cells, insulin, PDX1, MafA

## Abstract

Pancreatic beta cell function is an important component of glucose homeostasis. Here, we investigated the function of PIMT (PRIP-interacting protein with methyl transferase domain), a transcriptional co-activator binding protein, in the pancreatic beta cells. We observed that the protein levels of PIMT, along with key beta cell markers such as PDX1 (pancreatic and duodenal homeobox 1) and MafA (MAF bZIP transcription factor A), were reduced in the beta cells exposed to hyperglycemic and hyperlipidemic conditions. Consistently, PIMT levels were reduced in the pancreatic islets isolated from high fat diet (HFD)-fed mice. The RNA sequencing analysis of PIMT knockdown beta cells identified that the expression of key genes involved in insulin secretory pathway, *Ins1* (insulin 1), *Ins2* (insulin 2), *Kcnj11* (potassium inwardly-rectifying channel, subfamily J, member 11), *Kcnn1* (potassium calcium-activated channel subfamily N member 1), *Rab3a* (member RAS oncogene family), *Gnas* (GNAS complex locus), *Syt13* (synaptotagmin 13), *Pax6* (paired box 6), *Klf11* (Kruppel-Like Factor 11), and *Nr4a1* (nuclear receptor subfamily 4, group A, member 1) was attenuated due to PIMT depletion. PIMT ablation in the pancreatic beta cells and in the rat pancreatic islets led to decreased protein levels of PDX1 and MafA, resulting in the reduction in glucose-stimulated insulin secretion (GSIS). The results from the immunoprecipitation and ChIP experiments revealed the interaction of PIMT with PDX1 and MafA, and its recruitment to the insulin promoter, respectively. Importantly, PIMT ablation in beta cells resulted in the nuclear translocation of insulin. Surprisingly, forced expression of PIMT in beta cells abrogated GSIS, while *Ins1* and *Ins2* transcript levels were subtly enhanced. On the other hand, the expression of genes, PRIP/*Asc2/Ncoa6* (nuclear receptor coactivator 6), *Pax6*, *Kcnj11*, *Syt13*, *Stxbp1* (syntaxin binding protein 1), and *Snap25* (synaptosome associated protein 25) associated with insulin secretion, was significantly reduced, providing an explanation for the decreased GSIS upon PIMT overexpression. Our findings highlight the importance of PIMT in the regulation of insulin synthesis and secretion in beta cells.

## 1. Introduction

Glucolipotoxicity refers to the deleterious effects of chronically elevated glucose and fatty acid levels on beta cells. Although glucose and saturated fatty acid palmitate are acutely stimulatory, chronic exposure leads to impaired insulin synthesis, exocytosis, and cell death [1,2]. In the earlier stages of glucolipotoxicity (mild decompensation stage), a specific loss of glucose-stimulated insulin secretion (GSIS) is observed; however, insulin synthesis is protected [1]. In the subsequent stage (severe decompensation), beta cells are de-granulated and insulin synthesis is impaired, along with grossly deranged beta cell differentiation and alterations in the key metabolic and transcription factor gene expression. In the later stages (decompensation with structural damage and beta cell death), islets may lose their normal architecture along with the deposition of amyloid fibrils and enhanced beta cell death. Glucolipotoxicity causes the disruption of multiple signalling pathways, leading to beta cell dysfunction. At the molecular level, PDX1 (pancreatic and duodenal homeobox 1) function is impaired at multiple levels: mis-localization, reduced expression, DNA binding etc. [2]. Further, the DNA binding of MafA (MAF bZIP transcription factor A) and ATF6 (activating transcription factor 6) is also impaired. Furthermore, a long term oversupply of metabolic fuel results in the loss of differentiated phenotype along with the reduced expression of end-differentiated genes such as PDX1, MafA, etc. [1]. The results from knockout mice of transcription factors such as PDX1 [3], Nkx2.2 (NK2 homeobox 2) [4], Pax4 (paired box 4) [5], Neurog3 (neurogenin3) [6], and NeuroD1 (neuronal differentiation 1) [7] emphasise their importance in pancreatic development. The significance of transcription factors such as PDX1/IPF1 (MODY4) [8] and NeuroD1 (MODY6) [9] in pancreatic development was further reinforced by the analysis of mutations in patients with maturity-onset diabetes of the young (MODY). Transcription factors work in collaboration with the coactivator or corepressor proteins that modulate the transcriptional machinery to activate or repress gene transcription. PPARγ (peroxisome proliferator activated receptor gamma) coactivator 1 (PGC-1), a transcriptional coactivator of nuclear receptors (NRs), was shown to suppress β-cell energy metabolism and insulin release [10]. Further, PRIP (Peroxisome Proliferator-Activated Receptor (PPAR)-Interacting Protein), a PPAR gamma co-activator, regulates insulin secretion and β-cell mass [11].

PIMT (PRIP-Interacting protein with Methyl Transferase activity), a co-factor binding protein, is proposed to serve as a bridge between the HAT (Histone acetyltransferases) and non-HAT co-activator complexes assembled at the gene promoters [12]. It is also known by its synonym TGS1 (Trimethyl guanosine synthase1), which alludes to its function of 5′ m^7^G cap hypermethylation of a few RNA species such as small nuclear and nucleolar RNAs (snRNAs and snoRNAs), selenoprotein, and telomerase RNAs. PIMT/TGS1 is involved in pre-mRNA splicing, transcription, and ribosome biogenesis [13,14,15]. Using liver-specific PIMT knockout mice (PIMTΔLivKO mice), we earlier reported that glucose release was significantly reduced in PIMTΔLivKO hepatocytes as compared to PIMT^fl/fl^ hepatocytes, demonstrating that PIMT regulates hepatic glucose output [16]. We also showed that PIMT is up-regulated in rats fed a high-sucrose diet (HSD). HSD leads to chronic systemic inflammation, leading to elevated levels of circulating TNFα. The expression of PIMT is up-regulated upon the exposure of myoblasts or myotubes to TNFα, resulting in impaired insulin-stimulated glucose uptake [17]. Recent data showed that two-month-old β-cell-specific TGS1 knockout mice (βTGS1KO) exhibit hyperglycaemia, decreased serum insulin levels, and impaired glucose tolerance. Islets derived from these mice also show defective GSIS, indicating that TGS1 is crucial in adapting to insulin resistance [18].

Here, we investigated the role of PIMT (PRIP-Interacting protein with Methyl Transferase activity) in the regulation of insulin synthesis and secretion. We observed that PIMT levels were decreased in the islets of high fat diet (HFD)-fed mice. PIMT knockdown or overexpression in the beta cells led to defective glucose-stimulated insulin secretion (GSIS). Importantly, PIMT knockdown prompted the re-distribution of insulin to the nucleus of beta cells. A mechanistic analysis revealed the impact of PIMT on PDX1-driven insulin transcription and gene expression associated with insulin synthesis and secretion.

## 2. Results

### 2.1. PIMT (PRIP-Interacting Protein with Methyl Transferase Activity) Expression Is Reduced in Cells Treated with Glucolipotoxic Conditions and HFD-Fed Mice Islets

To study the effects of high glucose and palmitic acid in a cellular model, BRIN-BD11 pancreatic β-cells were treated with 25 mM of glucose and 300 µM of palmitic acid alone or in combination for 24 h, and the expression levels of PIMT and key β-cell markers, such as PDX1 and MafA (β-cell specific transcription factors), HDAC5 (Histone deacetylase 5), a corepressor, and GCK (Glucokinase), a glucose sensing enzyme, were examined by a qPCR and Western blot analysis. We observed that the protein levels of PIMT along with the mentioned key β-cell markers were significantly reduced under glucolipotoxic conditions (Figure 1A and Appendix A). The treatment of cells with high glucose alone did not alter PIMT, PDX1, and GCK protein levels (Figure 1A and Appendix A) but reduced the expression levels of MafA and HDAC5 (Figure 1A and Appendix A). On the other hand, the exposure of cells to palmitate alone or in combination with high glucose reduced the protein levels of PIMT, PDX1, MafA, HDAC5, and GCK (Figure 1A and Appendix A). Furthermore, the qPCR analysis revealed that the transcript levels of *PIMT/Tgs1*, *Pdx1*, *Hdac5*, *Ins1*, and *Ins2* were attenuated in cells challenged with glucolipotoxic stimuli (Figure 1B–F). Time-course experiments showed that PIMT protein level was enhanced after the treatment of cells with palmitate and glucose for 6 h, followed by a steady reduction up to 48 h (Figure 1G and Appendix A). Further, PDX1 protein levels were substantially reduced after the exposure of cells to glucose and palmitate for 24 h and 48 h (Figure 1H and Appendix A). In contrast, *Tgs1* transcript levels were unchanged up to 12 h, followed by a significant decrease post 24 h and 48 h treatment (Figure 1I). In parallel, we observed the progressive enhancement of the mRNA transcript levels of both *Ins1 (Insulin1)* and *Ins2 (Insulin2)* up to 12 h of treatment, followed by a marked reduction at 24 h and 48 h (Figure 1J,K).

Next, to study the effects of hyperglycaemia and hyperlipidaemia and its effects on the expression levels of PIMT and key markers of pancreatic β-cells in an in vivo type 2 diabetic animal model, we isolated islets from mice fed with a high fat diet (HFD) for a period of 16 weeks. HFD mice showed significant body weight gain, increased adipose tissue size, and enhanced levels of fasting sugar and blood insulin (Figure 1L). An analysis of the HFD-fed islets by Western blotting showed that the protein levels of PIMT and MafA, which maintain a mature beta cell phenotype, were repressed (Figure 1M).

### 2.2. RNA Sequencing Analysis of PIMT Knockdown Beta Cells Revealed Impairment of Insulin Secretory Pathway

Having observed that PIMT expression was altered in cells challenged with glucolipotoxic stimuli and the islets of HFD-fed mice, we sought to investigate the impact of PIMT on the global transcriptome of beta cells. An RNA sequencing analysis was performed under three different experimental conditions. Set 1: vehicle or glucose (25 mM) and palmitate (300 µM) treated control cells; Set 2: untreated control or PIMT knockdown cells; Set 3: control or PIMT knockdown cells treated with glucose (25 mM) and palmitate (300 µM), as depicted in (Figure 2A). Volcano plots for three conditions showed robust differential gene expression. At a cut-off of 2-fold change and FDR ≤ 0.05, 286, 382, and 1106 genes were significantly altered in Sets 1, 2, and 3, respectively (Figure 2B–D). Importantly, we observed that *Ins1* was significantly down-regulated, as depicted in the heat map of Set 2 (Figure 2F). A differential gene expression analysis also revealed alteration of key insulin synthesis and secretory pathway genes: *Pdx1* (key transcription factor of Ins genes); *MafA* (maintains mature beta cell phenotype); *Pax6* (paired box 6, maintains mature beta cell identity); *Klf11* (Kruppel-Like Factor 11: binds to the promoter of insulin); *Nr4a1* (Nuclear receptor subfamily 4, group A, member 1,important role in regulating beta cell proliferation), *Kcnj11* (Potassium inwardly-rectifying channel, subfamily J, member 11, depolarization of the membrane potential); *Kcnn1* (Potassium calcium-activated channel subfamily N member 1, repolarization of action potential); *Gnas* (GNAS complex locus, defines insulin secretory capacity of beta cells); *Rab3a* (member RAS oncogene family, GTP binding protein involved in the exocytosis of insulin granule); *Syt11* (Synaptotagmin11) and *Syt13* (Synaptotagmin13 calcium-insensitive synaptotagmins and regulators of the exocytosis of insulin from beta cells). Moreover, the pathway analysis showed that insulin secretory, insulin resistance, apoptosis, and mitophagy pathways were enriched in the top 20 pathways of the Set 2 experiments (Figure 2G). To predict the transcription factors that may be associated with PIMT co-activator binding protein in β-cells, we performed a transcription factor enrichment analysis upon PIMT knockdown using the Enrichr analysis tool. We observed that the transcription factors involved in beta cell differentiation (Nanog; Nanog homeobox, Klf4; Kruppel-Like Factor 11, Sox2; SRY-box transcription factor 2, Tcf3; transcription factor 3, and Pou5f; POU class 5 homeobox 1), energy metabolism (RXR; retinoid X receptor, LXR; nuclear receptor subfamily 1 group H member 3, PPAR alpha, PPARG), and transcriptional regulation in beta cells (PDX1, MafA, NeuroD1, Klf6; Kruppel-Like Factor 11, p300; E1A binding protein p300, CEBPB; CCAAT enhancer binding protein beta) were the main hits (Figure 2H).

### 2.3. PIMT Knockdown Attenuates Insulin Secretion and the Associated Transcriptional Machinery

Next, we validated selected differentially expressed genes by qPCR analysis. Here, we observed more than a 70% significant reduction in the mRNA levels of both *Ins1* and *Ins2* (Figure 3A–C). The mRNA levels of key transcription factors driving insulin synthesis, *Pdx1*, *Mafa*, and *Neurod1* were not changed upon PIMT knockdown (Figure 3D–F). However, the protein levels of PDX1, MafA, and NeuroD1 were diminished in PIMT ablated BRIN-BD11 cells (Figure 3G,H). Consistently, we found decreased protein levels of MafA and PDX1 in PIMT-depleted rat pancreatic islets (Figure 3I,J). Further, the transcript levels of insulin secretory pathway genes, such as *Kcnj11* (membrane potential depolarization) *Kcnn1* (membrane potential repolarization), *Syt13* (synaptotagmin 13, calcium-insensitive synaptotagmin regulator of insulin exocytosis), and *Gnas* (defines the insulin secretory capacity of beta cells), were significantly decreased in PIMT-deficient BRIN-BD11 cells (Figure 3K–N). Moreover, the gene expression of key transcription factors *Pax6* (insulin synthesis and secretion regulator), *Nr4a1*, and *Klf11* was also hampered upon PIMT ablation (Figure 3O–Q). The expression of *Rab3a*, *Syt11*, and *Gpi* (glucose-6-phosphate isomerase); *Gpr119*, G protein-coupled receptor 119, *Stxbp1*, and *Snap25* showed a subtle change (Figure 3R–W). Supporting transcriptomics and qPCR data, the protein levels of Kir6.2 (ATP sensitive K+ channel subunit Kir6.2) and synaptotagmin 13 were substantially reduced in the PIMT knockdown cells (Figure 3X).

Having observed the impact of PIMT on the mRNA levels of transcription factors driving insulin, the transcript levels of insulin, and the genes associated with insulin secretion, we next studied the functional role of PIMT in glucose-stimulated insulin secretion (GSIS). For this, BRIN-BD11 cells were nucleofected with either control shRNA or shPIMT and incubated for 72 h. Seventy-two hours of post nucleofection, the cells were subjected to a GSIS assay. Depletion of PIMT led to a ~1.5-fold reduction in insulin secretion (Figure 3Y), and the total insulin content in PIMT knockdown BRIN-BD cells was increased by ~20% (Figure 3Z and Appendix A).

### 2.4. PIMT Interacts with PDX1 and MafA and Is Recruited to Ins1 and Ins2 in Beta Cells

Next, we sought to investigate the recruitment of PIMT to insulin gene loci based on the two observations: (1) PIMT depletion led to repressed insulin transcript levels; (2) PDX1 that drives insulin transcription was enriched in the transcription factor analysis. In light of this, we performed ChIP (chromatin immunoprecipitation) experiments showing that PIMT was recruited to *Ins1* and *Ins2* promoters (Figure 4A). We quantified the recruitment by ChIP-qPCR analysis using the fold enrichment method and observed that PIMT was enriched ~four-fold at *Ins1* and *Ins2* promoters (Figure 4B,C). Primers used in the analysis targeted the PDX1, Mafa, and Neurod1 binding sites of *Ins1* and the PDX1-binding elements of *Ins2*, as depicted in Figure 4A. As PIMT was recruited to the PDX1-binding elements of the *Ins* genes, we sought to test if PIMT interacts with PDX1, MafA, and NeuroD1. For this, we performed co-immunoprecipitation experiments, which showed that PIMT basally interacted with PDX1 and MafA but not with NeuroD1 (Figure 4D–F).

### 2.5. PIMT Knockdown Prompted Localization of Insulin to the Nucleus in the Beta Cells

We studied the insulin localization in PIMT-ablated BRIN-BD11 cells by immunofluorescence. Interestingly, we observed that PIMT knockdown promoted nuclear localization of insulin (Figure 5A). Quantification of insulin and nuclear signals showed robust co-localization, as revealed by a high correlation coefficient (Figure 5C). While insulin was largely excluded from the nucleus in the control cells (Figure 5A,C), it partially co-localized with endoplasmic reticulum (ER; Figure 5B,D) in control shRNA-transfected cells. On the other hand, insulin was excluded from ER and was predominantly localized to the nucleus in PIMT-depleted cells. Insulin was observed in the nuclei of approximately 92% BRIN-BD11 cells upon PIMT ablation (Figure 5E). Moreover, we investigated the nuclear translocation of insulin in PIMT knockdown BRIN-BD cells by separating nuclear and cytoplasmic fractions, and insulin levels were significantly enriched by ~2.5-fold in the nuclear fraction of PIMT knockdown cells compared to the corresponding control (Figure 5F,G). To investigate the possible mechanism of translocation, we studied the expression of key genes involved in insulin translation and the processing of proinsulin. The mRNA levels of *Eif2ak3* (eukaryotic translation initiation factor 2 alpha kinase 3) increased by ~60%, while *Sec61a2* (SEC61 translocon subunit alpha 2) and *Stard10* (StAR-related lipid transfer domain containing 10) decreased by ~50%; moreover, a subtle change was observed in the mRNA levels of *Ide* (insulin degrading enzyme) (Figure 5H–M).

### 2.6. PIMT Overexpression Impairs Glucose-Stimulated Insulin Secretion and the Associated Gene Expression

Having observed that the ablation of PIMT reduced GSIS, we next sought to study the impact of PIMT overexpression on GSIS. For this, we ectopically expressed PIMT in BRIN-BD11 cells and subjected them to a GSIS assay. Contrary to our expectation, we observed that the forced expression of PIMT robustly suppressed GSIS (Figure 6A). Next, we studied the effect of PIMT on GLP-1 mediated insulin secretion using cAMP/CRE- Luc assay as a marker. For this, BRIN-BD11 cells were co-transfected with a PIMT overexpression construct, along with CRE- Luc plasmid, and treated with 10 µM of liraglutide (GLP-1 agonist). Consistent with the effects on GSIS, the overexpression of PIMT significantly blocked cAMP-driven CRE-Luc activity (Figure 6B). To understand how PIMT-forced expression attenuated insulin secretion, we probed the protein levels of key insulin gene transcription factors and regulators by Western blot. Consistent with the findings of PIMT knockdown experiments, we observed that MafA and PDX1 protein levels were increased by more than two-fold and 1.5-fold (Figure 6C and Appendix A), respectively. Further, the mRNA levels of *Pdx1* and *Mafa* and their transcriptional targets, *Ins1* and *Ins2*, were subtly increased upon PIMT overexpression (Figure 6D–H). Additionally, the levels of HDAC5, belonging to class IIa HDACs, were also enhanced (Figure 6B and Appendix A). Interestingly, forced expression of PIMT reduced the expression of PRIP (PPAR gamma-interacting protein), also known as *Asc2*, which interacts with PIMT and regulates the gene expression of target genes (Figure 6I). We also measured the mRNA levels of major secretory machinery genes: (i) *Stxbp1* (also known as Munc18-1): part of the SNAREs complex in the regulation of insulin exocytosis; (ii) *Syt13*: calcium-insensitive synaptotagmin co-localized with insulin in insulin granules; and (iii) *Snap25*: core protein of SNARE complex controlling insulin exocytosis from beta cells in response to stimulus. The expressions of *Stxbp1*, *Syt13*, and *Snap25* were reduced in PIMT overexpressing cells. The expression levels of *Stxbp1*, *Syt13*, and *Snap25* negatively correlated with the in vivo measurements of HbA1C and in vitro measurements of GSIS in human T2DM patients. Further silencing of *Syt13* led to diminished insulin secretion in human T2DM islets [19]. In our experiments, we observed that upon PIMT overexpression, the transcripts levels of *Stxbp1* (Munc18-1), *Snap25*, and *Syt13* were diminished by ~50% (Figure 6L–N). This may explain, at least in part, reduced GSIS in PIMT overexpressing cells. Moreover, we found that the mRNA levels of *Kcnj11*, a voltage-gated potassium channel also known as Kir6.2, which has an important role in GSIS, was decreased by ~20% (Figure 6K). Furthermore, the mRNA levels of *Pax6* were reduced by ~70% (Figure 6J). Pax6 has an important role in embryonic beta cell development; *Pax6* KO mice showed a decreased ATP/ADP ratio and reduced Ca^2+^ dynamics, both of which contribute to insulin secretion [20]. Protein levels of key secretory proteins SNAP25 (synaptosome associated protein 25) and synaptotagmin 13 (SYT13) were also significantly reduced upon PIMT overexpression (Figure 6O,P). The immunofluorescence analysis revealed insulin localization to the cytoplasm in both control and PIMT ectopically expressing cells (Figure 6Q,R), with a very low Pearson’s correlation coefficient for nuclear localization (Figure 6S).

## 3. Discussion

In summary, we reported the following main findings: (1) PIMT is a crucial regulator of insulin synthesis and secretion in the beta cells; (2) the deficiency of PIMT promotes the nuclear localisation of insulin; and (3), either the deficiency or overexpression of PIMT downregulates proteins involved in the insulin secretory pathway, resulting in decreased GSIS in the beta cells. The key findings are summarized in (Figure 7).

Chronic hyperglycaemia, chronic inflammatory stress, and hyperlipidaemia are the major drivers of defective β-cell function and beta cell death. Understanding the underlying molecular mechanisms of β-cell dysfunction is critical for the development of better therapies. In pursuit of this objective, we uncovered that PIMT is an important regulator of GSIS. Attesting to the importance of PIMT in β-cells, we observed that the exposure of β-cells to glucolipotoxic conditions significantly decreased both mRNA and protein levels of PIMT. Consistently, PIMT protein levels were also reduced in the islets harvested from HFD-fed mice.

The functional analysis revealed that PIMT regulates GSIS. The mechanistic analysis showed that PIMT controls the levels of key regulators of β-cells such as PDX1, MafA, NeuroD1, and HDAC5. PDX1 is the critical transcription factor for the differentiation of β-cells during embryogenesis: the deletion of PDX1 leads to pancreatic agenesis, and the partial loss of PDX1 in β-cells led to beta cell dysfunction and increased apoptosis through insulin/insulin-like growth factor pathway [3,21]. MafA-deficient mice develop abnormal glucose-stimulated insulin secretion as well as the abnormal architecture of islets, which affects the maturation of β-cells [22]. Supporting GSIS studies, the transcriptomics analysis highlighted the impact of PIMT on the transcript levels of *Ins1* and *Ins2*, genes associated with insulin secretory, mitophagy, and apoptosis pathways. Our results are consistent with the recent finding that β-cells-specific PIMT KO mice showed significantly decreased levels of insulin in the islet of Langerhans, as well as increased levels of ER stress markers *Atf4* (activating transcription factor 4), *Atf6* (activating transcription factor 6), *Eif2ak3* (eukaryotic translation initiation factor 2 alpha kinase 3), and *Chop* (C/EBP homologous protein) [18]. The importance of PIMT in β-cells and its impact on a diverse set of genes may be primarily attributed to its interaction with PDX1, a key transcription factor essential for pancreatic development and adult β-cell function. For example, using a combination of immunoprecipitation and ChIP experiments, we identified that PIMT interacts with PDX1 and MafA, and is recruited to the PDX1-binding elements of *Ins1* and *Ins2* genes. Further, the earlier findings that PIMT deletion leads to signs of ER stress and apoptosis may be partly due to its impact on PDX1 levels. Depletion of PIMT by RNAi attenuated the levels of PDX1, which controls the susceptibility of β-cells to ER stress-associated apoptosis [18,23]. The defects in PIMT-depleted cells may also stem from the decline in another key transcription factor, *Pax6*. Mutations of *Pax6* are associated with a diabetic phenotype and a drastic decrease in insulin positive cell number. Pax6 knockdown led to decreases in insulin cell content, insulin processing, and a specific defect of glucose-induced insulin secretion as well as a significant reduction in GLP-1 (glucagon like peptide-1 action in primary β-cells [24].

The observation that PIMT likely controls the re-localization of insulin to the nucleus is striking. The ablation of PIMT levels in β-cells led to the re-distribution of insulin from the cytoplasm/ER to the nucleus. However, the ectopic expression of PIMT showed no impact, suggesting that basal PIMT may control certain key players participating in the insulin translocation machinery. Insulin secretion by β-cells involve an orchestrated program of insulin biosynthesis, beginning with the preproinsulin mRNA, translation and translocation into the ER, proinsulin folding and export from the ER, and delivery via the Golgi complex to secretory granules for its maturation and ultimate storage. All of these steps are needed for the generation and maintenance of the total insulin granule pool, and defects in any of these steps may, weakly or strongly, perturb glycaemic control [25]. To the best of our knowledge, this is the first report that points to the translocation of insulin to the nucleus of insulin-producing β-cells. Earlier studies showed the nuclear translocation of insulin and probable mechanisms in the insulin-utilising cells [26]. Insulin may be translocated to the nucleus by two different mechanisms: (a) receptor-mediated endocytosis or (b) fluid phase endocytosis. Caveolae may serve as the main membrane sites where the signalling pathway cascade is triggered. Internalized insulin may be released from the endosomes into the cytoplasm, where it binds to cytosolic Ins-binding proteins (CIBPs) and escapes insulin-degrading enzyme (IDE)-mediated degradation. Subsequently, insulin is proposed to translocate through the nuclear pore into the nucleus where it associates with the nuclear matrix. Based on our current observations and earlier reports [18], we speculate that PIMT depletion results in ER stress, leading to preproinsulin aggregation and shuttling of the aggregates to the cytoplasm. The aggregates may be protected from IDE-mediated degradation and thereby prompt its association with CIBPs followed by nuclear import. In this direction, we analysed the transcript levels of the important genes associated with the regulation of translation, translocation to ER lumen, and subsequent processing of proinsulin in PIMT-ablated β-cells. While the transcript levels of IDE remained unchanged in the PIMT-depleted cells, the expression levels of (i) *Eif2ak3* (involved in ER stress-mediated translational suppression) [25]; (ii) *Sec61a2* (contributes to the transport of preproinsulin to ER lumen); and (iii) *Stard10* (required for glucose-stimulated Ca^2+^ dynamics, and insulin secretion and proinsulin processing) [27,28] were altered. Additional studies are needed to validate our speculations. Further, whether the nuclear re-distributed insulin is functional or not also requires investigation.

Either PIMT knockdown or PIMT overexpression led to decreased GSIS. Signalling defects in the insulin secretory pathway may partly explain such a discrepancy. PIMT ectopic expression resulted in the downregulation of the following genes associated with insulin secretion: (1) *kcnj11* (Kir6.2), *Kcnn1* (SK1), and GNAS (depolarization and repolarization of membrane potential; (2) *Rab3a*, *Stxbp1* (Syntaxin-binding protein 1), *Snap25*, and Synaptotagmins 11 and 13 (vesicular transport and membrane fusion). Further, Pax6, the key transcription factor, which regulates both insulin synthesis and the secretion of insulin, was also decreased. In contrast, the overexpression of PIMT increased HDAC5 (co-repressor) protein levels, which could exert a repressive effect on the secretory genes [29]. Moreover, we found that that Asc2/PRIP (PIMT interacting co-activator protein) was significantly decreased upon PIMT overexpression. A study by Yeom et al. found that PRIP expression was restricted to endocrine cells, and deletion of the PRIP led to decreased glucose-stimulated insulin secretion in mice [11]. Moreover, PIMT was discovered using PRIP/ASC2 as a bait in a yeast two-hybrid system [30]. Whether PIMT also interacts with PRIP in beta cells remains to be tested. Our findings suggest that the maintenance of PIMT levels in an optimum range is essential for effective GSIS. Either a reduction in the basal levels or increased expression of the transcriptional regulator PIMT may attenuate GSIS, albeit using different mechanisms. Such an effect is observed with another transcriptional regulator NR4A1 as well [31]. Taken together, the current study uncovers the importance of PIMT in the control of the GSIS of β-cells.

## 4. Materials and Methods

### 4.1. Cells Line and Animals

Pancreatic beta cell line BRIN-BD11 was used for the in vitro studies. The cells were cultured in the RPMI 1640 supplemented with 10% FBS and antibiotic and antimycotic (Anti-anti) from Himedia, Thane West, Maharashtra, India, (catalogue no. A002). Male Sprague–Dawley rats ranging from 200 to 250 g in body weight were housed with light and dark cycles. Commercially available chow diet pellet and water were used in the animal study.

### 4.2. High Fat Fed Mouse Model

Male C57BL/6 mice (6–7 weeks of age) were maintained on a high-fat diet with 60% kcal from fat (60% kcal fat in the form of lard, 20% kcal protein, 20% kcal carbohydrates; Cat # D12492, Research Diet, New Brunswick, NJ, USA) ad libitum with access to water for a period of 16 weeks to induce obesity. Animals maintained on regular chow diet were considered the control group. These experiments were approved by the Animal Ethics Committee of IICB, Kolkata (AEC NO. IICB/AEC/Meeting/May/2022/5). Disease induction was monitored and confirmed by the periodical measurement of body weight (once weekly). At the end of the induction period, plasma glucose and insulin were measured and islets were harvested.

### 4.3. Cell Lysis and Western Blotting

BRIN-BD11 cells were lysed using TENNS lysis buffer [120 mM of NaCl, 1% Triton X-100, 20 mM of Tris–HCl pH 7.5, 10% glycerol, 2 mM of EDTA and protease inhibitor cocktail (10 μg/mL each of aprotinin and leupeptin)]. Approximately 30 μg of protein was resolved by 10% SDS-PAGE and a Western blot analysis was carried out using specified antibodies. The catalogue and primary dilution of the antibodies used in the study are Anti-PIMT (Abcam, Cambridge, UK, catalogue no. Ab70559) 1:2000; anti-PDX1 (Cell Signaling Technologies, Danvers, MA, USA, catalogue no. 5679) 1:1000; anti-MafA (sc-390491) 1:200); anti-HDAC5 (sc-133225) 1:1000; anti-GCK (sc-17819) 1:200; anti-insulin (Cell Signaling Technologies, 8138) 1:1000; anti-Synaptotagmin13 (Novus Biologicals, NB2-20546) 1:1000; anti-SNAP25 (sc-390644 Santa Cruz Biotechnology, Finnell Street Dallas, TX, USA) 1:200; anti-Kir6.2 (sc-20809) 1:200; anti-NeuroD1 (Cell Signaling Technologies, 2833) 1:1000; and anti-GAPDH (Cell Signaling Technologies, 2118) 1:5000. The protein levels were normalized using GAPDH and Western blots were quantified by using ImageJ [32].

### 4.4. Real-Time PCR

Total RNA was isolated using TRIzol, and reverse transcription was performed using Superscript II First Strand cDNA Synthesis System, according to the manufacturer’s instructions. Twenty-microliter reactions were set up comprising 2 μg of total RNA, and 200 nM of random hexamers and oligo dT primers. qPCR reactions were performed using TB Green mix on QuantStudio 5 (Applied Biosystems, Waltham, MA, USA). Data were normalized to GAPDH using the Ct method (^ΔΔ^Ct). The cDNAs from the respective samples were diluted at 1:5 for the target genes and 1:50 for reference genes (18 s, actin, and gapdh). Relative gene expression was plotted considering the control sample as 1. The gene IDs of the genes used in the qPCR are shown in Appendix A.

### 4.5. Cloning of shRNA and Generation of Stable PIMT Knockdown Cells

shRNA was designed targeting the CDS region and cloned into the pLKO3.1 vector at EcoR I and Age I sites. Transfection of shRNA was achieved by a nucleofection device using an in-house developed electroporation solution [33]. Briefly, cells were transferred to a sterile 0.2 cm cuvette and electroporated using the nucleofection programme for the LnCap cell line. After nucleofection, the cells were gently resuspended in 1 mL of a pre-warmed RPMI medium. All cells were seeded in 6-well plates and grown at 37 °C and 5% CO_2_. The medium was replaced with complete RPMI medium the following day and incubated for 72 h. Post-incubation, cells were selected using puromycin at (2.5 µg/µL).

### 4.6. RNA Sequencing and Data Analysis

RNA quality and quantity was measured by using Bioanalyzer (Agilent Technologies, Stevens Creek Blvd Santa Clara, CA, USA). An RNA integrity number (RIN) greater than 8.0 was used for library construction. The total RNA (150 ng) was utilized for generating libraries for RNA sequencing. The RNA was fragmented and then converted to cDNA according to the literature protocol. The cDNA was end-repaired and further purified using Ampure XP beads. The cleaned cDNA was adapter ligated and purified using Ampure XP beads. These adapters ligated fragments were then subjected to 12 cycles of PCR using the primers provided in the kit. The PCR products were purified using Ampure XP beads. The quantification and size distribution of the prepared library was determined using Qubit fluorimeter and Agilent Tape station D1000 Kit (Agilent Technologies), according to the manufacturer’s instructions.

The transcriptome libraries were constructed using the NEB adapters and were sequenced on Illumina HiSeq at a 150 nt read length using paired-end chemistry. The raw data obtained were then processed for the low-quality bases and adapters contamination. The raw reads were subjected to contamination (structural RNA/low complexity sequences) removal by mapping with bowtie 2-2.2.1. The decontaminated data set was mapped to the *Rattus norvegicus* reference sequence. Reads mapping to the gene list were counted using the feature count module of the sub reads package. The read counts were normalized in DESeq2-3.5 and subjected to a differential expression analysis. Significantly up and downregulated genes (differentially expressed) were selected based on 2-fold change with a *p* value of <0.05 (FDR: 0.05; Student *t*-test, unpaired). Hierarchical clustering was performed with the programs Cluster (uncentered correlation; average linkage clustering) and Tree view (Eisen et al., 1998). Term enrichments of differentially regulated genes were calculated based on mouse genome database as background. GO terms with corrected *p* < 0.05 were considered to be significantly enriched for the pathway analysis. For the ‘Enrichr’ analysis, differentially altered genes (criteria: cut-off: fold change > 1.5 and FDR: 0.05) were submitted to the ‘Enrichr’ tool. The tool used the KEGG pathway 2022 human pathway to analyse the possible disease conditions in all the sets and the possible transcription factors involved in the gene alterations using the ChIP-X enrichment analysis (ChEA) gene set library. The disease pathways and enriched transcription factors were calculated based on the following terms: (1) −log10 *p* values that are computed based on alterations in the number of genes that are known primary or secondary targets of transcription factors; (2) odds ratio that indicates a measure of association between input and outcome; and (3) combined score that signifies deviation from the expected rank and is calculated by log(*p*) × z, where *p* is the *p* value and z = deviation. The significantly differentially expressed gene in Sets 1, 2, and 3 are shown in Appendix A respectively.

### 4.7. Islet Isolation

The resected rat and mouse pancreatic tissues (control and HFD-fed) were collected in RPMI 1640 nutrient medium (catalogue no. AT120 from Himedia) and immediately processed for islet cell isolation. Pancreatic tissue was freed of muscle and adipose tissues and washed 3 times with RPMI 1640 nutrient medium. The tissue was then subjected to mechanical chopping and digestion with collagenase D (Sigma-Aldrich, St. Louis, MO, USA; 1 mg/mL) in HBSS buffer (CaCl_2_ 12.6 mM, MgCl_2_·6H_2_O 4.9 mM, KCl 53.3 mM, KH_2_PO_4_ 4.4 mM, NaCl 1379.3 mM, NaHPO_4_·7H_2_O 3.4 mM, D-glucose 55.6 mM) at 37 °C for 15 to 20 min. The resultant dispersed pancreatic islets were washed 3 times with cold 1× HBSS buffer. After isolation, islets were cultured for 12 h in 11 mM of glucose RPMI 1640 medium (containing 100 IU/mL of penicillin, 100 g/mL of streptomycin, and 10% foetal bovine serum) before being handpicked and cultured at 37 °C with 5% CO_2_ and saturated humidity. Islets from control and high fat diet-fed (HFD) mice were pooled from three animals. Protein was extracted in RIPA lysis buffer and quantified by Bradford.

### 4.8. Lentivirus Generation and Transduction

The HEK 293T cells were seeded in a 10 cm dish at 70% confluence and transfected with pPax2, pMD2G, and shPIMT plasmids by polyethylenimine (catalogue no. 24765-1 from polysciences, Valley Road Warrington, USA). After seventy hours, virus supernatant was collected and concentrated by overlaying on a 10% sucrose-containing buffer (50 mM of Tris-HCl, pH 7.4, 100 mM of NaCl, 0.5 mM of ethylene diamine tetra acetic acid [EDTA]) at a 4:1 *v*/*v* ratio and centrifuged at 10,000× *g* for 4 h at 4 °C [34]. Lentiviral transduction of shPIMT for the knockdown of PIMT was used to infect the islets. Approximately 300 islets were handpicked for each group. To access the core of islets, they were trypsinized (0.5×) for 3 min and incubated with the concentrated virus in serum-free media for 12 h, then subsequently transferred to complete media (RPMI with 10% FBS) [35]. Seventy-two hours post-transduction, the islets were harvested for protein extraction. Protein was extracted with RIPA lysis buffer. Western blotting was performed with 30 µg of protein and probed with anti-PIMT (catalogue no. 70559 from Abcam) and anti-MafA (catalogue no. 390491 from Santa Cruz Biotechnology).

### 4.9. GSIS, ELISA, and CRE-Luc Assay

The cells were seeded in a 60 mm dish for the transfection of PIMT (3 μg DNA) by using Lipofectamine 2000 at a 1:2 ratio of DNA to lipofectamine (catalogue no. 11668019). Twenty-four hours’ post-transfection, the cells were used for glucose-stimulated insulin secretion (GSIS). Briefly, the cells were washed twice with KRB buffer 0.1% BSA and pre-incubated at 37 °C for 60 min in 250 μL of Krebs-Ringer bicarbonate (KRB) buffer (115 mM of NaCl, 4.7 mM of KCl, 1.28 mM of CaCl_2_, 1.2 mM of KH_2_PO_4_, 1.2 mM of MgSO_4_, 10 mM of NaHCO_3_, 0.1% (wt./vol) fatty acid free BSA, pH 7.4). The cells were washed once with glucose-free KRBH and then incubated for 30 min with KRBH containing 16.7 mM of glucose. The supernatants were collected for insulin determination, and the protein content in cell lysates was measured by Bradford assay to normalize the insulin values of the corresponding cell supernatants. Insulin was measured by rat insulin radioimmunoassay (rat insulin ELISA from Mercodia, Sylveniusgatan 8A, SE-754 50, Uppsala, Sweden, catalogue no. 10-1250-01) following the manufacturer’s instructions. cAMP-responsive element-luciferase reporter plasmid (encoding the luciferase reporter gene under the control of the minimal promoter and three tandem repeats of CRE transcriptional response element; CRE3X-Luc) and SFB-PIMT vector were co-transfected with Lipofectamine 2000. Four hours’ post-transfection, the cells were transferred to 12-well plates at a density of 0.1 million cells per well. After 24 h, the cells were treated with and without an appropriate GLP-1R (glucagon like peptide 1 receptor) agonist liraglutide from (Victoza) at a concentration of 10 µM in complete medium for another 4 h. The medium was then aspirated, the cells were lysed, and luciferase activity was measured.

### 4.10. Immunoprecipitation and ChIP (Chromatin Immunoprecipitation)

For the immunoprecipitation experiments, precleared 500 μg cell lysate was incubated with PIMT antibody (4 μg) overnight at 4 °C. Post-incubation, the beads were washed with lysis buffer followed by boiling with Laemmli buffer. The enriched samples were separated and probed with the indicated antibodies. For the ChIP experiments, the cells were fixed with 1% (*w*/*v*) formaldehyde for 10 min at room temperature, followed by one wash using ice-cold PBS. The fixation reaction was stopped with the addition of 2 M of glycine solution for 15 min. The cells were scraped from the flasks with 10 mL of ice-cold PBS and centrifuged for 5 min at 1000 rpm, after which they were lysed using SDS ChIP lysis buffer. The chromatin obtained was sheared by sonication at an amplitude of 45% with a 10 s on/off pulse for 2 min in ice. The optimized sonication conditions resulted in 200–1000 bp DNA fragments. Post-lysis, 250 μL of the isolated chromatin was diluted three times using ChIP dilution buffer and incubated with 3 μg of PIMT antibody or IgG antibody at 4 °C overnight.

### 4.11. Immunofluorescence

Stable PIMT knockdown BRIN-BD11 cells were grown on a coverslip under puromycin selection. ER was visualized by transfecting the cells with mcherryER3 (mCherry-ER-3 Addgene plasmid # 55041). The cells were then washed in phosphate buffer saline (PBS) and incubated at 37 °C for the desired time in complete medium, after which they were fixed in 3% paraformaldehyde. Subsequently, the cells were permeabilized with 0.1% TritonX100 and incubated with insulin monoclonal antibody overnight (1:200; Cell Signaling Technology, catalogue no. L6B10). The next day, the cells were washed and incubated with secondary antibody conjugated with Alexa fluor 492 (Thermo Scientific, Waltham, MA, USA, catalogue no. A32731) and mounted in Vectashield mounting medium (Vector Laboratories) containing DAPI. Images were quantified for co-localization study by using ImageJ, version 1.52o [36] and two different individuals scored the number of cells showing nuclear or cytoplasm localization of insulin.

### 4.12. Nuclear and Cytoplasmic Fractionation

The cells were collected in PBS and pelleted. Cell pellets were lysed in 50 mM of Tris-Cl. pH 6.8, 150 mM of Nacl, and 0.1% Nonidet P-40, with protease inhibitor. The cells were triturated 5 times with a P1000 micropipette. A fraction of this was collected as whole-cells lysate; the remaining lysate was centrifuged at 19,000× *g* for 15–30 s, and the supernatant was collected and labelled as cytosolic fraction. The pellet was washed with lysis buffer two times, lysed in 2× Laemmli buffer, and labelled as the nuclear fraction. Whole-cell fraction and nuclear fraction were sonicated before boiling the samples [37].

## Figures and Tables

**Figure 1 ijms-24-08084-f001:**
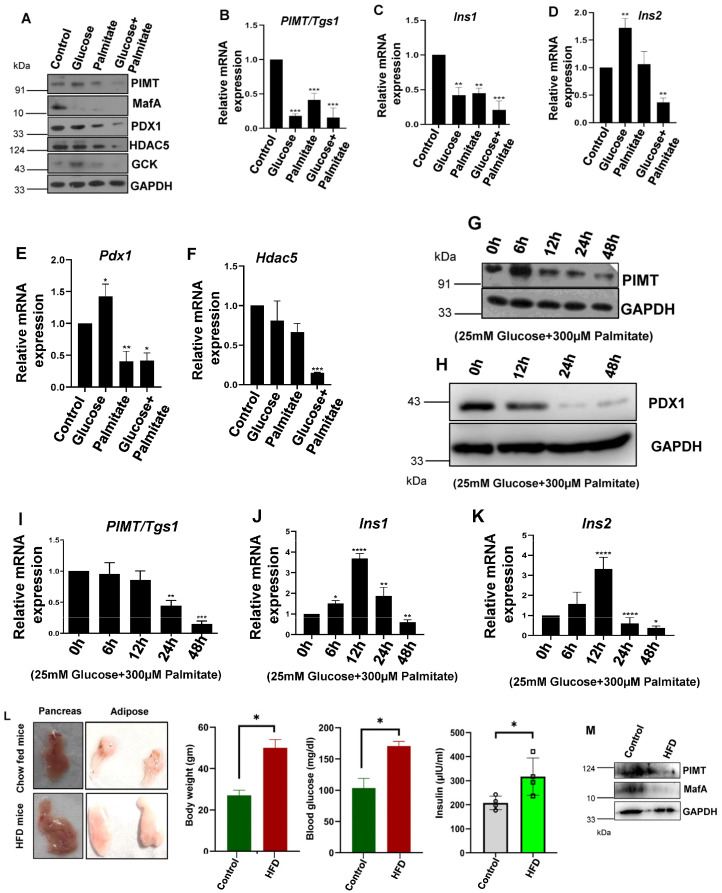
PIMT (PRIP-Interacting protein with Methyl Transferase activity) levels were decreased in BRIN-BD11 cells exposed to glucolipotoxicity and HFD-fed mice islets. (**A**–**F**) BRIN-BD11 cells were treated with glucose (25 mM) or palmitic acid (300 μM) alone or in combination. Levels of PIMT and key beta cell markers were analysed by Western blotting (**A**) and qPCR (**B**–**F**). (**G**–**J**) Analysis of the expression levels of PIMT and other beta cell markers in BRIN-BD11 cells exposed to glucose (25 mM) and palmitic acid (300 μM) for the indicated time-points by Western blotting (**G**,**H**) and qPCR (**I**–**K**). (**L**) Phenotypic assessment of pancreas and adipose tissues from chow- and HFD-fed mice. Side panels: Body weight, serum levels of glucose and insulin in chow and HFD-fed mice, (n = 3). (**M**) Analysis of PIMT protein levels in HFD-fed islets (pooled islets from three control and three HFD mice pancreas) by Western blotting. Data are representative of three independent experiments. Numerical data are shown as mean ± SD. Statistical analysis was performed using Tukey’s multiple comparison test. A * *p* < 0.05; **, *p* < 0.01; ***, *p* < 0.001; ****, *p* < 0.0001 value was considered statistically significant.

**Figure 2 ijms-24-08084-f002:**
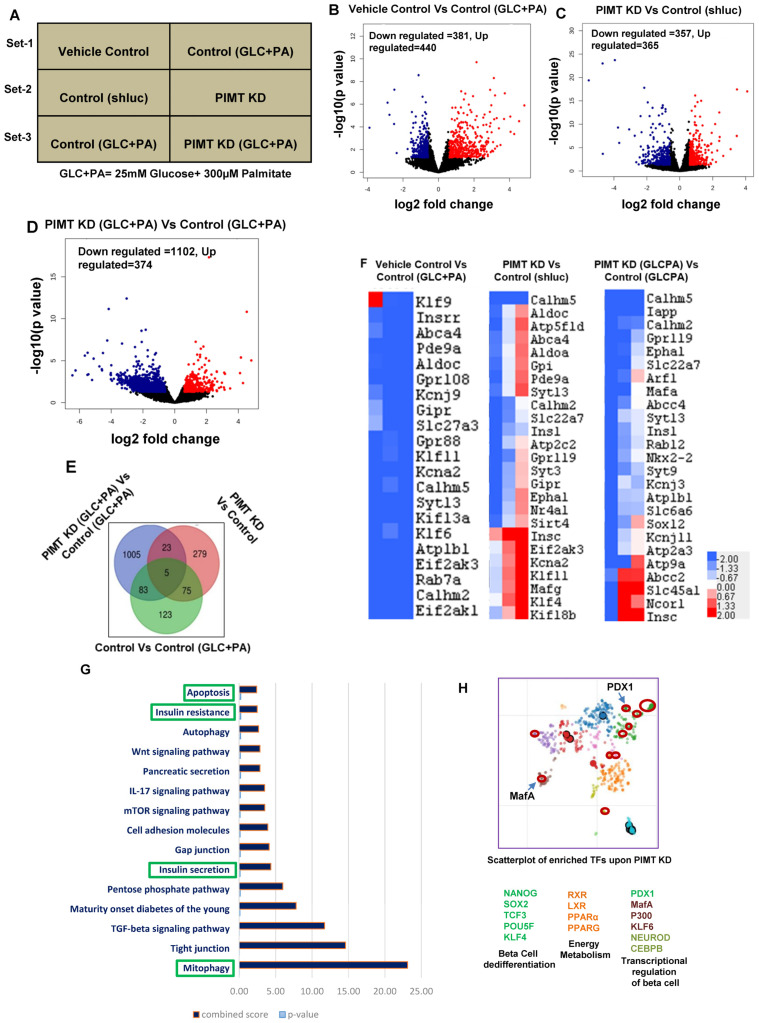
RNA sequencing analysis of PIMT knockdown beta challenged with glucolipotoxic conditions. (**A**) Representation of experimental conditions—Set 1: vehicle treated versus 25 mM glucose and 300 µM palmitate treated cells; Set 2: control knockdown cells versus PIMT knockdown cells; Set 3: control knockdown treated vs. PIMT knockdown treated cells. (**B**–**D**) Volcano plot analysis of Set 1, 2, and 3 genes. (**E**) Venn diagram showing commonly affected genes in the three sets. (**F**) Heatmap analysis of insulin secretory pathway of Sets 1, 2, and 3. (**G**) The top 25 affected pathways in Set 2 were identified using the Enrichr tool by submitting the differentially altered genes from Set 2 (fold change > 1.5, FDR < 0.05). The KEGG 2022 human pathways library was used to analyse the possible disease conditions. (**H**) A scatterplot of enriched transcription factors in Set 2 (cut-off: FC > 1.5, FDR < 0.05) was analysed with the “Enrichr” tool to identify the possible transcription factors involved in the gene alterations using the ChIP-X enrichment analysis (ChEA) gene set library (the brighter the colour, the greater the significance of the transcription factor). Data are representative of three independent experiments.

**Figure 3 ijms-24-08084-f003:**
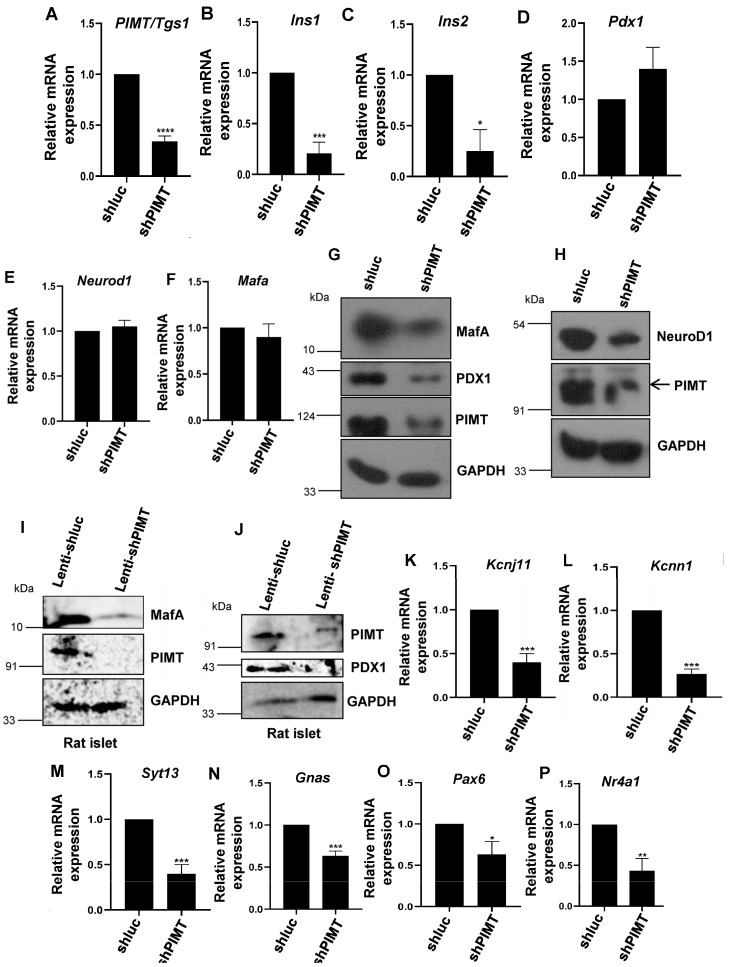
PIMT regulates *Ins1* and *Ins2* expression. (**A**–**F**) qPCR analysis of PIMT (**A**), *Ins1* (**B**), *Ins2* (**C**), and other beta cell markers (**D**–**F**) in PIMT knockdown beta cells. (**G**–**J**,**X**,**Z**) Western blot analysis of PIMT knockdown BRIN-BD11 cells (**G**,**H**,**X**,**Z**) or rat pancreatic islets (**I**,**J**) with the indicated antibodies. (**K**–**W**) qPCR analysis of selected insulin secretion-associated genes in PIMT knockdown BRIN-BD11 cells. (**Y**) Insulin secretion in PIMT knockdown cells stimulated with different concentrations of glucose. Data are representative of three independent experiments. Numerical data are shown as mean ± SD. Statistical analysis was performed using Student’s *t*-test. A * *p* < 0.05; **, *p* < 0.01; ***, *p* < 0.001; ****, *p* < 0.0001 value was considered statistically significant.

**Figure 4 ijms-24-08084-f004:**
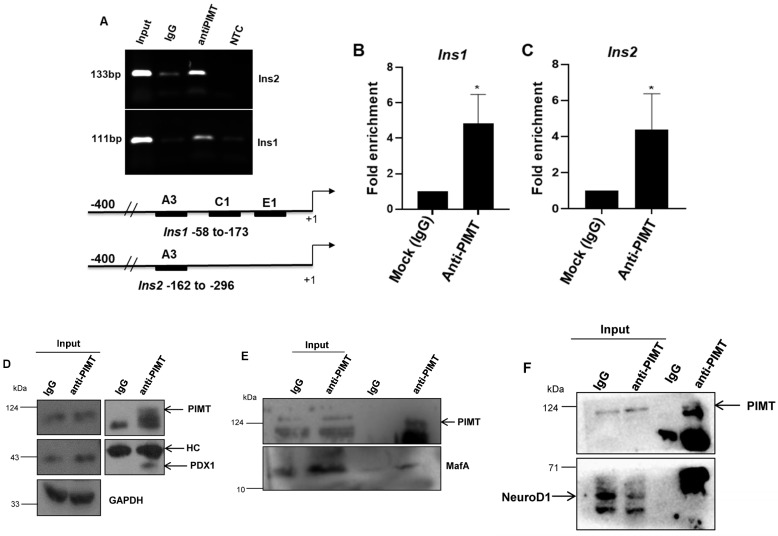
PIMT interacts with PDX1 and MafA is recruited to *Ins1* and *Ins2* promoters. (**A**) ChIP analysis of BRIN-BD11 cells using PIMT (upper panel). Lower panel: schematic depicting the binding elements of PDX1 on *Ins1* and *Ins2* genes. (**B**,**C**) Fold enrichment-based qPCR analysis of chromatin immunoprecipitates of the samples analysed in (**A**). (**D**–**F**) Immunoprecipitation analysis of BRIN-BD11 cells with the indicated antibodies. HC: heavy chain. Data are representative of three independent experiments. Numerical data are shown as mean ± SD. Statistical analysis was performed using Student’s *t*-test. A * *p* value < 0.05 was considered statistically significant.

**Figure 5 ijms-24-08084-f005:**
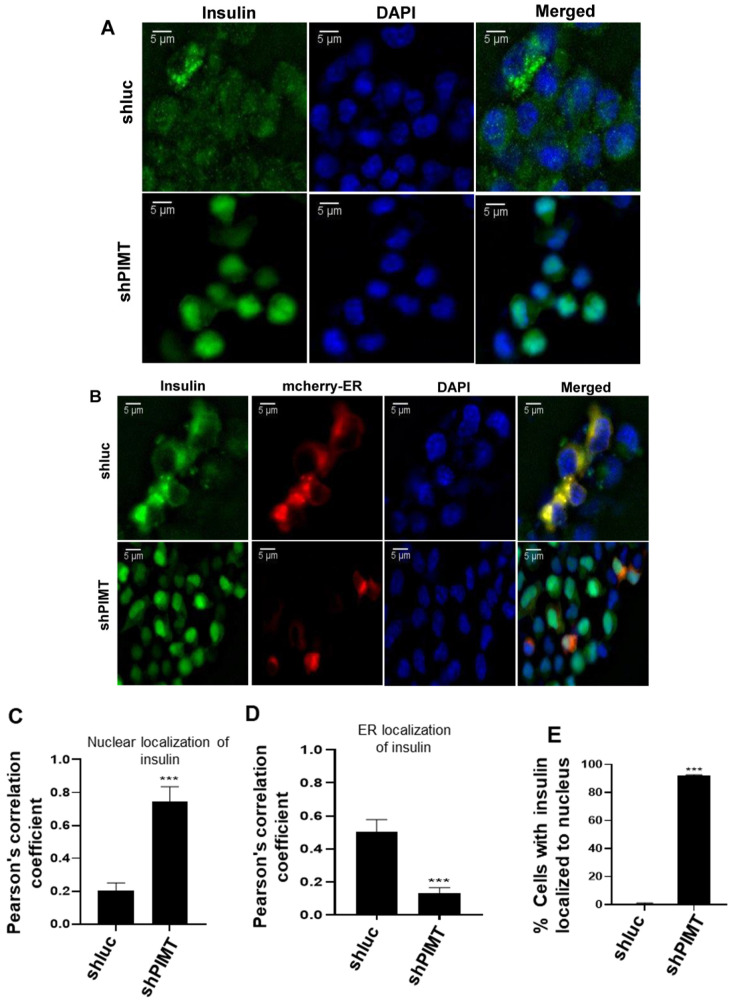
Loss of PIMT promotes nuclear translocation of insulin. (**A**–**D**) Immunofluorescence analysis of insulin in PIMT knockdown BRIN-BD11 cells along with nuclear staining by DAPI (**A**,**C**) or ER staining using mCherry-ER reporter (**B**,**D**) scale bar represents 5 µM. Quantification of co-localization of insulin with nucleus (**C**) or ER (**D**) by ImageJ analysis. Insulin: green, ER: red, nucleus: blue. (**E**) Percentage of cells with nuclear localization of insulin. (**F**,**G**) Probing insulin in nuclear and cytoplasmic fractions of PIMT knockdown beta cells with the indicated antibodies (**F**) and the quantification of insulin in the nuclear fraction (**G**). (**H**–**M**) Analysis of mRNA levels of *Eif2ak3* (eukaryotic translation initiation factor 2 alpha kinase 3), *Sec61a1* (SEC61 translocon subunit alpha 1), *Sec61a2* (SEC61 translocon subunit alpha 2), *Stard10* (StAR-related lipid transfer domain containing 10), and *Ide* (insulin degrading enzyme) in PIMT knockdown cells by qPCR. Data are representative of three independent experiments. Numerical data are shown as mean ± SD. Statistical analysis was performed using Student’s *t*-test. A * *p* < 0.05; **, *p* < 0.01; ***, *p* < 0.001 value was considered statistically significant.

**Figure 6 ijms-24-08084-f006:**
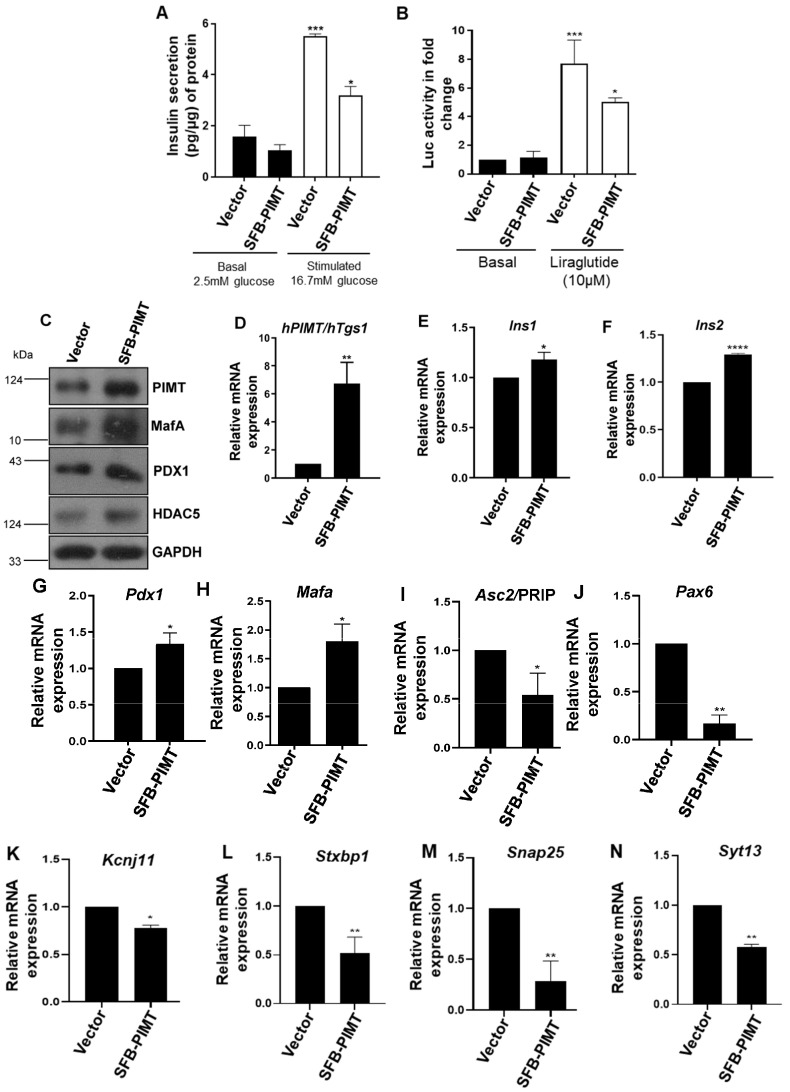
PIMT overexpression decreased GSIS and key insulin secretory pathway genes. (**A**) Insulin secretion in PIMT-overexpressing BRIN-BD11 cells at different glucose concentrations. (**B**) cAMP-driven CRE-Luc reporter activity in PIMT-overexpressing BRIN-BD11 cells in response to 10 µM of liraglutide. (**C**) Analysis of the protein levels of PDX1 and MafA, and HDAC5, in PIMT-overexpressing BRIN-BD11 cells. (**D**–**N**) mRNA levels of PIMT/*Tgs1*, *Ins1*, *Ins2*, *Pdx1*, *Mafa*, and genes associated with insulin secretion in PIMT-overexpressing BRIN-BD11 cells. (**O**,**P**) Immunoblotting analysis of PIMT-overexpressing BRIN-BD11 cells with SNAP25 (synaptosome associated protein 25) (**N**) and SYT13 (synaptotagmin 13) (**O**) antibodies. (**Q**–**S**) Immunofluorescence analysis of the insulin in PIMT-overexpressing BRIN-BD11 cells, scale bar represents 5 µM (**Q**), percent of cells with insulin localized to cytoplasm (**R**) and co-localization quantification (**S**). Data are representative of three independent experiments. Numerical data are shown as mean ± SD. Statistical analysis was performed using Student’s *t*-test. A * *p* <0.05, ** *p* < 0.01; *** *p* < 0.001; **** *p* < 0.0001 was considered statistically significant.

**Figure 7 ijms-24-08084-f007:**
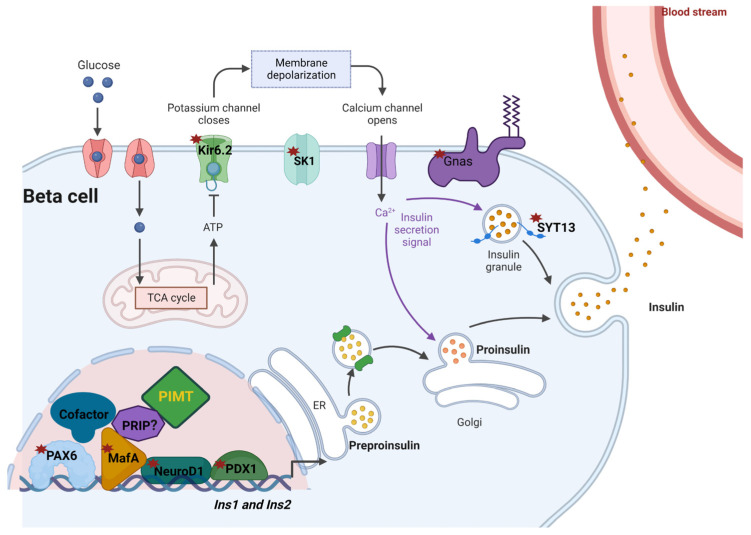
Impact of PIMT on insulin synthesis and secretion machinery. PIMT is recruited to the insulin promoter and interacts with PDX1 and MafA to regulate insulin transcription. PIMT deficiency leads to a reduction in the levels of transcription factors and insulin secretion-associated genes such as PDX1, MafA NeuroD1, Pax6, Gnas, Klf11, Nr4a1, Kir6.2, SK1 channel, and Synaptotagmin 13 (highlighted by asterisk).

## Data Availability

RNA-sequencing processed data, identifying DEGs are available on Mendeley data at https://data.mendeley.com/drafts/yjpfhgn23k (accessed on 5 January 2023).

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
