# Peer review of "PIMT Controls Insulin Synthesis and Secretion through PDX1"

_ijms, 2023, doi:10.3390/ijms24098084_

Round 1
Reviewer 1 Report
In this manuscript, Sharma et al. describe role of PIMT in pancreatic beta-cells. They show that PIMT expression is reduced in BRIN-BD11 cells treated with glucolipotoxic conditions and in HFD-fed mice islets. PIMT knockdown in BRIN-BD11 cells and rat islets results in reduced mRNA expression and impaired glucose-stimulated secretion of insulin, and reduced protein level of MAFA, NEUROD1 and PDX1. Paradoxically, mRNA expression of Mafa, Neurod1 and Pdx1 is not changed in these cells and islets. PIMT interacts with PDX1 and is recruited to Ins1 and Ins2 promoters. They also demonstrate that PIMT overexpression impairs glucose-stimulated insulin secretion.
Results are clearly described with careful examination. However, this manuscript presents several experimental data that differ from conventional consensus and previously published results. In addition, there are lots of typos and some experiments are not described in sufficient detail. These concerns should be addressed before further evaluation of this study.
Major comments:
1. Fig. 1: Results in this study show reduced expression of PIMT in BRIN-BD11 cells treated with glucolipotoxic conditions and HFD-fed mice islets. Mafa expression is also reduced in the islets of HFD-fed mice. However, TSG1 KO mice study (J Biol Chem 298, 2022) revealed that TGS1 is up-regulated in mice islets exposed to a high-fat diet and in human β-cells from type 2 diabetes donors. Mafa expression is not changed in the islets of HFD fed-mice in many other published studies. These are important points of this study and should be examined at a molecular level.
2. Fig. 3: 70% reduction in expression of PIMT results in normal mRNA expression of Mafa, Neurod1 and Pdx1, but reduced protein expression of MAFA, NEUROD1 and PDX1. Please show molecular mechanism of this phenomenon. Are promoters of these genes inactivated in these cells?
3. Fig. 5: As authors mention, nuclear translocation of insulin has not been shown so far and thus should be confirmed. Can some blocking peptides for the insulin antibody inhibit the nuclear signals after immunocytochemistry? Can subcellular fractionation of these cells also show nuclear translocation? In addition, please show data that can explain molecular mechanism for translocation of insulin.
Minor points:
4. Authors should seek the assistance of an English-proficient colleague or professional manuscript editing service to improve the clarity of the manuscript.
5. Please clarify biological replicates of each experiment in all the figures.
6. Fig. 6E-G: mRNA level of Mafa should be added to these figures.
7. Results of Fig. S1 should be included in Fig. 1.
8. Lines 147-157: Expression of these genes should be shown as independent figures.
9. It seems that “Fig. 3F” should be “Fig. 2F” (lines 149), “Fig. 6B” (line 294 and 297) should be “Fig. 6C”. “2.6” is missing in line 280. There is no Figure 7 in this manuscript (line 349).
Reviewer 2 Report
The authors highlighted the importance of PIMT in the regulation of insulin synthesis and secretion in beta cells. They showed that PIMT levels were decreased in the islets of high fat diet (HFD) fed mice. It is intriguing that both PIMT knockdown and over-expression in beta cells led to defective glucose stimulated insulin secretion. They attributed the effect on PIMT on insulin transcription and gene expression associated with insulin release and synthesis via PDX1. If PIMT controls insulin synthesis and secretion through PDX1, you will expect the overexpression of PIMT to increase GSIS via PDX1 but that is not the case .The evidence provided in this manuscript with regards to mechanisms regulating insulin secretion by PIMT is not sufficient.
Major comments:
11. Does Pdx1 mRNA or protein levels decreases in time-depended manner when BRIN-BD11 cells are exposed to both glucose and palmitic acid similar to PIMT as shown in fig 1G and 1H
22. I understand that working with islets is difficulty but the protein bands for PIMT and MAFA for Fig. 1K is not clear and kindly repeat with extra experiments (pulling islets from 3 different animals for one western blot experiment does not make N=3, instead it is N=1)
33. Authors should provide the list of genes significantly altered in sets 1,2 and 3 as depicted in Figure 2B-D as supplementary tables.
44. Does PIMT knockdown affect insulin content since it is well established that Knockdown of Pdx1 in beta cell reduce insulin content? Please provide absolute values for glucose-stimulated insulin secretion (GSIS), insulin content as well as insulin secretion expressed as percentage of content for fig. 3M and 6A
55. Does PIMT interacts with MAFA or NEUROD1
66. What is the effect of PIMT knockdown on STXBP1 and SNAP25 .
Minor comments :
77. Kindly provide catalogue and dilutions of all primary antibodies used for western blot in the method section (line 456)
88. Figures for western blot should include MW for each protein detected.
99. Provide a better image for Heatmap analysis for sets 1, 2,3 with clear labels (Fig 2F: the gene names are running into each other)
110. Provide the whole blots of the three independent experiments for Fig. 3G-J as well as Fig.3L, Fig. 6C, 6N and 6O with clear Molecular Weight (you have only provided the blots used in the figures- correct me if I am wrong)
111. Include all the gene ID (catalogue from Applied Biosystems) used in the qpCR experiment in the method section (line 463).
Author Response
Please see the attachement

Round 2
Reviewer 1 Report
Authors addressed most of concerns raised by this reviewer.
Reviewer 2 Report
The manuscript is much improved but the following needs to be addressed:
Point 2 : Biological replication is very important when it comes to scientific studies, I hope you can address this.
Point 4: The claim that PIMT controls insulin synthesis and secretion through PDX1 is not entirely true since :
1. You have no validated PIMT also interact with MAFA
2. Knockdown of PIMT enhances insulin content but PDX1 knockdown reduces insulin content.
3. PIMT over-expression robustly suppress GSIS but leads to 1.5 fold increment of PDX1 protein level (you will expect improvement in GSIS with increase PDX1?)
Point 10: The quality of blot from the western blot experiments is very low standard. Since most of the findings of this manuscript is based on this experiment, I suggest that you redo some of them:
Ø For instead in Biological replicates of Figure 3G and H, in some experiments you have multiply bands for shPIMT cells for the detection of PIMT but in others you have single band ( N1 vs N2 vs N6). How do you explain this?
Ø Kindly provide quantification of the blot after normalization with control in the figure and mark bands used in the figure in the raw western blot data.
Ø Did you have 2 biological replicates for PDX1 for Figure 6C? because the bands for N=1 and N=3 for PDX1 is not visible?
Ø Did you have 2 biological replicates for SYT13 for Figure 6P? I see only N=1 and N=2?
Point 11 : Kindly provide all primers designed for qPCR experiments for reproducibility purposes.